Tracing RNA viruses associated with Nudibranchia gastropods

http://orcid.org/0000-0003-0685-1618 Rosani Umberto umberto.rosani@unipd.it
Department of Biology, University of Padova , Padova , Italy
Breitbart Mya
Electronic publication date: 2022 May 13
Publication date: 2022
Volume: 10
Electronic Location ID: e13410
Received 2022 Feb 10; Accepted 2022 Apr 19
Copyright: © 2022 Rosani
Copyright year: 2022
Copyright holder: Rosani
License: This is an open access article distributed under the terms of the Creative Commons Attribution License, which permits unrestricted use, distribution, reproduction and adaptation in any medium and for any purpose provided that it is properly attributed. For attribution, the original author(s), title, publication source (PeerJ) and either DOI or URL of the article must be cited.
License URL: https://creativecommons.org/licenses/by/4.0/

Keywords: RNA viruses, Nudibranchia, RNA-seq, Virome, Marine biodiversity

Funding: Italian National Project PRIN2017 Umberto Rosani was supported by the Italian National Project PRIN2017 (Viral diversity and impacts on deep-sea biodiversity and ecosystem functioning, VIRIDE). The funders had no role in study design, data collection and analysis, decision to publish, or preparation of the manuscript.

==============================
Background

Nudibranchia is an under-studied taxonomic group of gastropods, including more than 3,000 species with colourful and extravagant body shapes and peculiar predatory and defensive strategies. Although symbiosis with bacteria has been reported, no data are available for the nudibranch microbiome nor regarding viruses possibly associated with these geographically widespread species.

Methods

Based on 47 available RNA sequencing datasets including more than two billion reads of 35 nudibranch species, a meta-transcriptome assembly was constructed. Taxonomic searches with DIAMOND, RNA-dependent-RNA-polymerase identification with palmscan and viral hallmark genes identification by VirSorter2 in combination with CheckV were applied to identify genuine viral genomes, which were then annotated using CAT.

Results

A total of 20 viral genomes were identified as bona fide viruses, among 552 putative viral contigs resembling both RNA viruses of the Negarnaviricota, Pisuviricota, Kitrinoviricota phyla and actively transcribing DNA viruses of the Cossaviricota and Nucleocytoviricota phyla. The 20 commonly identified viruses showed similarity with RNA viruses identified in other RNA-seq experiments and can be putatively associated with bacteria, plant and arthropod hosts by co-occurence analysis. The RNA samples having the highest viral abundances showed a heterogenous and mostly sample-specific distribution of the identified viruses, suggesting that nudibranchs possess diversified and mostly unknown viral communities.

Introduction

With an estimated 108 viral particles per millilitre of coastal water, oceans can be referred as hubs of viral diversity (Suttle, 2007; Mushegian, 2020). Here, viruses of unicellular organisms can govern the fate of host populations and influence global environmental cycles (Suttle, 2007; Brum et al., 2015; Sunagawa et al., 2020). As viruses are obligate intracellular parasites, their diversity reflects host genetic variety within the three domains of life (Krehenwinkel, Pomerantz & Prost, 2019). In this context, host-virus combinations frequently show co-adaptation processes, although viruses infecting multiple hosts and host-jumps have often been reported for invertebrate viruses (Shi et al., 2016). Completely leveraging the high-throughput power of modern sequencers, shotgun metagenomics is contributing to untangle the fascinating complexity of host and virus diversity and co-associations in the marine environment (Edwards & Rohwer, 2005; Carlos, Castro & Ottoboni, 2014; Kodzius & Gojobori, 2015). Starting from seawater filtrates or sediments subjected to RNA/DNA purification, amplification and subsequent high-throughput sequencing (HTS), viral metagenomics aims to reconstruct metagenome assembled genomes (MAGs) (von Meijenfeldt et al., 2019), often showing low similarity with database entries (Sullivan, 2015; Shi et al., 2016; Sunagawa et al., 2020). Although is itself challenging, the identification of viruses from metagenomic data is only an introduction to the understanding of host-to-virus associations and, finally, to the characterization of the functional roles of viral communities, also called viromes (Sullivan, 2015; Mara et al., 2020). Hence, both the distribution and the ecosystemic importance of viruses are probably underestimated and, to some extent, undiscovered (Koonin & Dolja, 2013; Garmaeva et al., 2019).

Differently from environmental metagenomics, the study of organism-associated biomes (holobiomes) can provide an overview of organism–organism associations (Hardoim et al., 2021) together with a snapshot of local viromes, as shown for the human virome (Garmaeva et al., 2019; Koonin, Dolja & Krupovic, 2021). Perhaps this approach is suffering less from limitations relating to proper host assignation, although additional challenges should be considered. Primarily, the low abundance of viruses within a healthy organism, together with the small size of viral compared to host genomes, makes their identification puzzling (Roux et al., 2015). Notably, some marine organisms can act as a magnifying glass for local viromes, due to their water filtration habits (Rosani et al., 2019). This has been exploited to reconstruct mimivirus genomes from oysters, as well as tracing human faecal footprints from mussel samples (Andrade et al., 2015; Olalemi et al., 2016). Although challenging, untangling host-associated viromes can also provide hints to trace the distribution of pathogenic viruses and to study host antiviral responses in non-model species (Rosani & Gerdol, 2017; Waldron, Stone & Obbard, 2018; Rosani et al., 2019).

Despite genomics, and with it metagenomics, having come on leaps and bounds over the past decades, several marine taxonomic groups have been overlooked resulting in a partial representation of the biological diversity of the sea (Greninger, 2018). One example pertains to the taxonomic group of gastropod nudibranchs, which includes more than 3,000 species seldom represented in sequence databases. These animals are characterized by colourful and varied body shapes, with sizes in the millimetric to centimetric range (Dean & Prinsep, 2017; Goodheart et al., 2018). Nudibranchs are known to derive defensive metabolites from the sponges they eat (Cheney et al., 2016), with several nudibranch-associated compounds that have been functionally studied highlighting their bioactivity against microbes and even viruses (Mudianta et al., 2016; Dean & Prinsep, 2017; Kristiana et al., 2020; Avila, 2020). Noticeably, viral-or bacterial-mediated diseases are not known for nudibranchs, whereas a fungus-like protist of the phylum Labyrinthulomycota has been associated with yellow and brown spot diseases observed in Tritonia diomedea (Collier et al., 2017). Infected nudibranchs mount an effective response able to encapsulate the pathogen with a lower impact on the fitness of the organism (McLean & Porter, 1982). In the context of global changes, nudibranchs have been proposes as sentinels of oceans. In particular warm-adapted nudibranch species appeared more susceptible to thermal changes, making these species highly vulnerable in the current temperature increasing scenario (Armstrong, Tanner & Stillman, 2019). RNA sequencing (RNA-seq) data are rare for these species, and most of them have been used to study the nervous system (Senatore, Edirisinghe & Katz, 2015), to trace nudibranch–gastropod phylogeny (Goodheart et al., 2017) or for comparative morphology studies (Goodheart et al., 2018). Noticing the complete absence of information regarding nudibranch-associated viruses, I exploited the transcriptomic datasets of 35 nudibranch species and, by performing a meta-transcriptomic analysis, I applied different pipelines to identify viruses associated with these species.

Materials and Methods

Data retrieval

The SRA archive (https://www.ncbi.nlm.nih.gov/sra) was interrogated on the 1st of December 2021, to retrieve 47 Nudibranchia transcriptomic datasets. RNA-seq samples characterized by paired-read layout, random or PCR selection (no size selection), based on Illumina technology were selected (Table 1). The NCBI protein (nr) databases were downloaded in December 2021, together with the annotation databases used for the tools described below (VirSorter2, CheckV and CAT).

Table 1 Short-read archive analyzed in the present work.

BioProject	Organism	Run ID	No. of reads (M)	Sampling location	Library type	
PRJNA252890	Tritonia tetraquetra	SRR1721590	128.7	Canada	PolyA	
PRJNA270545	Hermissenda crassicornis	SRR1719366	105.6	USA	RT-PCR	
PRJNA279852	Antiopella barbarensis	SRR1950942	45.2	USA	cDNA	
PRJNA279852	Austraeolis stearnsi	SRR1950943	42.5	USA	cDNA	
PRJNA279852	Berghia stephanieae	SRR1950951	46.7	USA	cDNA	
PRJNA279852	Catriona columbiana	SRR1950949	52.1	USA	cDNA	
PRJNA279852	Dendronotus venustus	SRR1950948	54.4	USA	cDNA	
PRJNA279852	Dirona picta	SRR1950946	50.3	USA	cDNA	
PRJNA279852	Doto lancei	SRR1950945	48.9	USA	cDNA	
PRJNA279852	Favorinus auritulus	SRR1950950	51.5	USA	cDNA	
PRJNA279852	Flabellinopsis iodinea	SRR1950940	60.6	USA	cDNA	
PRJNA279852	Hermissenda opalescens	SRR1950939	50.4	USA	cDNA	
PRJNA279852	Melibe leonina	SRR1950947	49.5	USA	cDNA	
PRJNA279852	Nanuca occidentalis	SRR1950953	60.9	USA	cDNA	
PRJNA279852	Palisa papillata	SRR1950952	44.6	USA	cDNA	
PRJNA279852	Trinchesia albocrusta	SRR1950944	61.7	USA	cDNA	
PRJNA279852	Tritonia festiva	SRR1950941	51.5	USA	cDNA	
PRJNA279852	Tritoniopsis frydis	SRR1950954	52.1	USA	cDNA	
PRJNA282347	Tritonia tetraquetra	SRR2004329	56.1	Australia	cDNA	
PRJNA319376	Aeolidiella alba	SRR3726702	39.2	French Polynesia	RANDOM	
PRJNA319376	Anteaeolidiella chromosoma	SRR3726695	42.5	Mexico	RANDOM	
PRJNA319376	Bornella anguilla	SRR3726697	29.8	Australia	RANDOM	
PRJNA319376	Dermatobranchus	SRR3726698	37.3	Australia	RANDOM	
PRJNA319376	Eubranchus rustyus	SRR3726692	43.8	USA	RANDOM	
PRJNA319376	Hancockia uncinata	SRR3726694	42.5	United Kingdom	RANDOM	
PRJNA319376	Learchis evelinae	SRR3726693	38.0	USA	RANDOM	
PRJNA319376	Limenandra confusa	SRR3726703	38.0	French Polynesia	RANDOM	
PRJNA319376	Lomanotus vermiformis	SRR3726706	46.0	Panama	RANDOM	
PRJNA319376	Nanuca parguerensis	SRR3726707	39.8	Panama	RANDOM	
PRJNA319376	Noumeaella rubrofasciata	SRR3726700	36.3	USA	RANDOM	
PRJNA319376	Phidiana lynceus	SRR3726705	38.1	Panama	RANDOM	
PRJNA319376	Scyllaea fulva	SRR3726701	42.1	French Polynesia	RANDOM	
PRJNA319376	Spurilla braziliana	SRR3726704	40.1	Panama	RANDOM	
PRJNA319376	Tenellia	SRR3726699	41.0	Indonesia	RANDOM	
PRJNA319376	Unidentia angelvaldesi	SRR3726696	36.0	Australia	RANDOM	
PRJNA327379	Melibe leonina	SRR3738852	118.8	USA	PolyA	
PRJNA342152	Tritonicula hamnerorum	SRR4190242	48.3	USA	PolyA	
PRJNA420367	Melibe leonina	SRR6333767	58.5	USA	RANDOM	
PRJNA440245	Tritonia tetraquetra	SRR6875319	12.8	Canada	RT-PCR	
PRJNA440245	Tritonia tetraquetra	SRR6875318	13.7	Canada	RT-PCR	
PRJNA440245	Tritonia tetraquetra	SRR6875317	13.7	Canada	RT-PCR	
PRJNA440245	Tritonia tetraquetra	SRR6875316	12.9	Canada	RT-PCR	
PRJNA440245	Tritonia tetraquetra	SRR6875315	12.4	Canada	RT-PCR	
PRJNA445612	Hermissenda crassicornis	SRR6894132	14.1	USA	RT-PCR	
PRJNA445612	Hermissenda crassicornis	SRR6894131	12.9	USA	RT-PCR	
PRJNA445612	Hermissenda crassicornis	SRR6894130	13.7	USA	RT-PCR	
PRJNA445612	Hermissenda crassicornis	SRR6894129	15.1	USA	RT-PCR	
Note:

BioProject ID, organism, run ID, size (no. of reads), collection date, location and library selection method were reported for the 47 selected samples.

Transcriptome assembly and preliminary analysis

RNA-seq datasets were trimmed for quality and to remove sequencing adaptors using fastp with default parameters (Chen et al., 2018). Trimmed reads were de novo assembled per sample using the CLC assembler (CLC Genomic Workbench v.20; Qiagen, Germantown, MD, USA), setting bubble size and word size graph parameters to automatic and the minimum allowed contig length to 200 bp. The cd-hit tool (Fu et al., 2012) was used to reduce the redundancy of the contigs, applying a cut-off of 99% similarity.

Taxonomic annotation of contigs and identification of viruses

A global taxonomic annotation of the RNA contigs was performed using DIAMOND (Buchfink, Xie & Huson, 2015) blastx searches against the NCBI nr database. The DIAMOND output file was meganized and uploaded in MEGAN6 for visualization (Huson et al., 2007). To properly identify viral contigs, two different approaches were applied. Firstly, to identify the footprint of the viral RNA-dependent-RNA-polymerases, palmscan (Babaian & Edgar, 2021) was applied with the more conservative parameters on the whole collection of RNA contigs. Secondly, to identify viral contigs based on viral hallmark genes, VirSorter2 (Guo et al., 2021) in combination with CheckV v0.8.1 (Nayfach et al., 2021) were used. VirSorter2 was run with a cut-off of 0.5 to maximize sensitivity, applying 1.5 kb as the minimal required length and searching for ‘RNA viruses’. The RNA sequences were used as input for coding sequence (CDS) prediction with prodigal v2.6.3 (Hyatt et al., 2010), which were then annotated against Pfam (release 32.0) and a custom viral HMM database to detect viral sequences. Putative viral contigs were then passed through CheckV to identify possible host genes and handle duplicate segments of circular contigs. Contigs with at least one identified viral gene or with a VirSorter2 score > 0.95 or a hallmark gene count > 2 (the last two parameters were applied in the absence of viral genes in a given contig) were considered as viral. All the putative viral contigs were annotated using CAT v5.0.4 (von Meijenfeldt et al., 2019), which used as input the proteins predicted by prodigal.

Viral distribution analysis

To evaluate the distribution of the retrieved viruses among the collection of datasets in the NCB SRA database, the RNA-dependent-RNA-polymerase regions obtained from the core-virome were extracted and used as query for palmID v.0.04 in the Serratus open-science viral discovery platform (Edgar et al., 2022). Briefly, for each query the tool provided a quality evaluation based on the conservation of the three functional motifs of the RdRP (motif A, B and C) and plotted a confidence value on the distribution of values obtained from 15,010 canonical viral RdRP from GenBank. Moreover, a word cloud analysis based on the STAT k-mer analysis (Katz et al., 2021) performed on the samples including similar viruses was considered to identify possible viral hosts.

Results

Taxonomic characterization of nudibranch meta-transcriptome

The search for the term ‘Nudibranchia’ in the nucleotide, protein, short-read (SRA) and bibliographic (PubMed) NCBI databases resulted in a limited number of hits upfront to 1,947 taxonomic entries at NCBI. Most of the 135 SRA entries referred to RNA-seq experiments based on Illumina technology, whereas no one genome draft is available for these species. Among these SRA samples, 47 datasets pertaining to 35 nudibranch species and accounting for 2.01 billion reads were selected (Table 1). De novo meta-transcriptome reconstruction, based on per-sample assemblies, resulted in 3,078,050 contigs, reduced to 2,929,504 contigs after the removal of highly similar sequences (>99%). Taxonomic classification based on blastx searches against the NCBI database assigned a match to 28.8% of these RNA sequences, with most of them referring to Mollusca (60%), and in particular to Aplysia spp. (100 k contigs), which represented the nearest genome-sequenced taxonomic entry within the Euthyneura clade. Notably, a considerable number of contigs (105 k) were assigned to Symbiodinium spp., a known symbiotic alga of marine metazoans including corals and molluscs. Together with a relatively small number of bacterial (11 k), fungal (1 k) and Deuterostomia hits (14.2 k), 453 contigs referring to viruses were identified.

Identification of the nudibranch-associated viruses

To identify bona fide viruses associated with these nudibranch RNA samples, DIAMOND blastx results were implemented with two different approaches: either used to identify the footprint of the viral RdRPs (palmscan) or to identify viral hallmark genes based on curated viral HMM databases. A total of 37 high-confidence RdRP footprints were identified by palmscan, whereas 128 putative viral contigs were identified by VirSorter2 (Table S1). Including the contigs identified by DIAMOND, a total of 552 contigs were classified as possible viruses, with only 19 contigs commonly identified by all the three methods (Fig. 1A, Data S1). The size distribution of the viral contigs increased from blastx to Virsorter2 results with the 19 common contigs placed in the upper part of the distribution (Figs. 1B and 1C). A taxonomic annotation of the 552 contigs based on the predicted CDS was performed with CAT, revealing that palmscan identified only contigs classified as ‘RNA virues’, whereas Virsorter2 identified also hits similar to metazoan genomes (Table S2). Like palmscan, Virsorter2 identified RNA viruses with the single exception represented by a Cossaviridae hit (ssDNA virus) identified due to an ORF encoding a parvovirus non-structural protein NS1 domain (SRR1950949_18608 in Data S1). Overall, several hits matching the phyla Kitrinoviricota, Negarnaviricota and Pisuviricota were identified, together with abundant unclassified viruses (Fig. 2 and Table S2). Regarding the 19 shared contigs plus the one identified by palmscan and Virsorter2, four contigs showed similarity to Beihai picorna-like virus 75 (Table 2). These contigs resembled three partial and one complete viral genome and their similarity to Beihai picorna-like virus 75 (KX883381) ranged from 46% to 53%. Two other contigs showed similarity to Wenzhou picorna-like virus 46, whereas four contigs were similar to different viruses belonging to the Pisuviricota phylum (Table 2).

Figure 1 Nudibranch-associated viruses.

(A) Comparison between the number of viral contigs identified by DIAMOND blastx, palmscan and Virsorter2 pipelines. (B) Length distribution of the retrieved contigs by the different approaches plus the 19 shared contigs. (C) Representation of the 19 shared viral genomes plus one genome identified by palmscan and Virsorter2 only. For each genome, the positions of the identified ORFs (white arrows) and of the predicted PFAM domains (black arrows) were reported.

Figure 2 Taxonomic classification of 552 viral contigs performed by CAT.

The annotations of the contigs classified as ‘Viruses’ were counted at the phylum level (NC; not classified).

Table 2 Annotation of 20 viruses associated with nudibranch samples.

Contig ID	Size (bp)	NCBI nt database (DIAMOND)	Phylogenetic classification (CAT)	Reported host	Suggested host
(palmID)	
Description	Phylum	Order	Family	Species	
SRR3726693_23	6,219	Riboviria sp. isolate 3nj-RDRP-3				Beihai picorna-like virus 75	Echinoderm	Metagenome	
SRR1950953_7108	9,998	Riboviria sp. isolate 3nj-RDRP-3				Beihai picorna-like virus 75	Echinoderm	Metagenome	
SRR1950940_818	8,834	Beihai picorna-like virus 72	Pisuviricota			Beihai picorna-like virus 72	Anthozoa	Bacteria (Fig. 3A)	
SRR1950953_2265	13,015					Beihai charybdis crab virus 1	Crustacea	Gastropod (Fig. 3B)	
SRR1950953_444	10,163	Beihai picorna-like virus 75				Beihai picorna-like virus 75	Echinoderm	Metagenome	
SRR3726693_131	3,162	Riboviria sp. isolate 3nj-RDRP-3				Beihai picorna-like virus 75	Echinoderm	Metagenome	
SRR1950953_981	11,238					Wenzhou picorna-like virus 46	Pomacea canaliculata	Metagenome (viral - porifera)	
SRR1950949_2406	9,654							Bacteria (Fig. 3C)	
SRR3726693_286	10,990					Wuhan snail virus 2	Mollusca		
SRR3726693_3931	9,200					Wenzhou picorna-like virus 46	Pomacea canaliculata	Metagenome (viral - porifera)	
SRR3726693_6245	6,102	Beihai picorna-like virus 56	Pisuviricota			Beihai picorna-like virus 56		n.d.	
SRR1950942_4074	8,133	Wenzhou picorna-like virus 19	Pisuviricota			Wenzhou picorna-like virus 19	Gastropoda	Plant (Fig. 3D)	
SRR1950951_5889	6,244	Basavirus sp. isolate BaV/21164						Bacteria (Fig. 3E)	
SRR3726705_1238	3,321							Bacteria (Fig. 3F)	
SRR3726699_38607	4,326	Eptesicus fuscus rhabdovirus							
SRR1950948_28126	3,196	Barns Ness breadcrumb sponge hepe-like 1	Kitrinoviricota	Hepelivirales	Hepeviridae	Barns Ness hepe-like virus 1	Porifera	Arthropoda (Fig. 3H)	
SRR1950948_38692	1,519	Barns Ness breadcrumb sponge hepe-like 2	Kitrinoviricota	Hepelivirales	Hepeviridae	Barns Ness hepe-like virus 2	Porifera		
SRR3726694_24261	3,010		Negarnaviricota	Mononegavirales	Nyamiviridae				
SRR3726696_22111	4,656	Biomphalaria virus 4	Pisuviricota	Picornavirales		Biomphalaria virus 4	Biomphalaria	Bacteria (Fig. 3H)	
SRR3726698_7138	1,701	Hubei earwig virus 1					Dermaptera		
Note:

For each contig, identified by the contig ID and size, the annotation obtained using blastn against the NCBI nt database (description) and the taxonomic classification obtained using CAT (phylum, order, family and species) were reported together with the suggested host of the best hit or by palmID. The viral contigs were ordered by total number of mapped reads. Underlined hits referred to genomes including low-confidence RdRPs (no palmID analysis was performed). Bolded taxonomic classifications referred to information not retrieved by CAT but added by palmID. SRR3726698_7138 represented the hit identified by palmscan and Virsorter2 only (annotations was retrieved using blastx).

Nudibranch-associated RNA virus distribution among the ‘planetary RNA virome’

A total of 15 high-confidence RdRP proteins were used to trace the distribution of the corresponding viruses within the collection of SRA datasets by using palmID implemented into the Serratus open science platform (Edgar et al., 2022). Association analysis based on the STAT kmer analysis performed on the RNA-seq datasets including similar viruses was used to retrieve possible host-virus associations (Fig. 3). The most abundant nudibranch-associated virus, Beihai picorna-like virus 75 (SRR3726693_23), was identified in metagenome-derived datasets poorly supporting possible hosts (Table 2). SRR1950953_226, which showed low similarity with a virus associated with crustacea (Beihai crab virus 1), was associated with arthropods by palmID. Differently, Wenzhou picorna-like virus 19 (SRR1950942_4074), initially associated with gastropods, is likely a plant virus. Viruses similar to Barns Ness hepe-like virus 1 and Barns Ness hepe-like virus 2, initially identified as possible porifera-associated viruses (Waldron, Stone & Obbard, 2018) but also found associated with bivalves (Rosani et al., 2019), revealed a possible association with arthropods. Beihai picorna-like virus 72 (SRR1950940_818) is possibly associated with Bacteria, as Biomphalaria virus 4 (SRR3726696_22111) and the unclassified SRR1950951_5889, SRR3726705_1238 and SRR1950949_2406 viruses. Notably, the two viruses similar to Wenzhou picorna-like virus 46 (SRR1950953_981 and SRR3726693_3931) were associated with a viral metagenome sample obtained from porifera (Rhopaloeides odorabile) as well as to several Pomacea canaliculata samples.

Figure 3 Word cloud graphs depicting taxonomic cooccurrences in the SRA samples including sequences similar to the identified viruses.

The letters referred to the last column of Table 2.

To further investigate possible host–virus associations in specific samples, a taxonomic classification of the assembled contigs of the eight samples including more than 500 viral reads was performed separately (Fig. S1). Except for nudibranchs, poor evidence of possible hosts of shared viruses was detected in independent samples (e.g. SRR1950953 and SRR3726693), nor for the highly abundant viruses found in sample SRR3726693, although bacteria are present in all these samples and might represent the genuine viral hosts (Fig. S1). An evident contamination of Dinophyceae (i.e. Symbiodinium spp.) is detectable in samples SRR1950951 and SRR3726699, but no viruses are shared between these samples. Low-level contaminations of Anthozoa are detectable in all samples, although the low abundance rejects the possibility that these species can act as hosts of the most abundant viruses. Differently, sample SRR1950953 showed Hydrozoa contamination, possibly representing the host of SRR1950953_226 and SRR1950953_981 viruses.

Discussion

Gastropod nudibranchs are intriguing models to investigate host–microbe associations and symbiosis from a functional point of view (Cheney et al., 2016; Mudianta et al., 2016; Kristiana et al., 2020). Notably, nudibranch-associated microbes are poorly known, and viruses have never being reported nor mentioned in the scientific literature. Besides the general interest of tracing viruses associated with these organisms, this will be preliminary to the understanding of possible roles mediated by viruses among nudibranch symbiosis as well as to untangle nudibranch antiviral defences. So far, no viral- or bacterial-mediated diseases are known in these species, possibly suggesting the presence of effective antiviral and antibacterial defences.

Here, the identification of viruses was approached by alternative and partially complimentary methods. Taxonomic classification of contigs (blastx) against the whole NCBI nr database identified both putative DNA and RNA viruses, although suffering from the poor representation of nudibranch species among databases. This aspect introduced abundant false positives in the identification of viruses, namely host contigs claimed as viral.

Differently, both palmscan and Virsorter2 genuinely identified RNA viruses. The former is designed to identify the footprint of the viral RdRPs (Edgar et al., 2022), an exclusive feature of RNA viruses. The latter identified viral hallmark genes and also provide a qualitative evaluation on the retrieved genomes (CheckV), reducing the identification of incomplete viruses, such as DNA viruses using RNA data. Overall, the sum of these methods identified 552 viral contigs, whereas the combination identified a core set of 19 viruses only. Sporadic DNA viruses identified in few samples and showing similarity with Herpesviruses, Mimiviruses or Poxviruses were discharged due to their genomic incompleteness.

Arguably, this result revealed the importance of using complimentary methods in order to detect high-confidence viruses associated with selected biological sample (e.g. nudibranchs), differentiating this analysis from a mere explorative survey focused on the detection of viruses. Likewise, although computationally very performant, the search of the RdRP only identified few viruses. Dedicated tools such as Virsroter2 limits false positives and increase the sensitivity by targeting multiple viral hallmark genes instead of the RdRP only, although requiring more computational effort.

By the search of the core nudibranch-virome among the ‘planetary RNA virome’ (Edgar et al., 2022), similar hits were retrieved as well as putative host-pathogen associations based on co-occurrence analysis.

Together, these results highlighted that nudibranchs possess different viromes, showing some similarities to viruses previously identified in gastropods, like Wenzhou picorna-like virus 46 and Wenzhou picorna-like virus 19 (Shi et al., 2016). Interestingly, the most represented virus showed similarity to Beihai picorna-like virus 75, initially found associated to echinoderms (Shi et al., 2016), although also present in several metagenomic datasets. Massive coverage in a single sample, together with the absence of evident signs of contamination of other possible host species, suggested that these viruses are hosted by nudibranchs (Learchis evelinae), and their role might be take into consideration in future research. Differently, all the other viruses are characterized by a lower coverage and co-occurrences analysis revealed that they possibly resembled bacterial viruses, which are still poorly studied (Wolf et al., 2020).

Indeed, nudibranch RNA samples are characterized by ‘physiological contaminations’, probably due to their feeding habits (corals) or mutualistic associations, here depicted by contigs of Symbiodinium spp. (Burghardt & Wägele, 2014; Mies et al., 2017). In this context, the host of the SRR1950940_818 contig similar to Beihai picorna-like virus 72 might be a coral, confirming its initial attribution to Anthozoa (Shi et al., 2016), although palmID associated also this virus with bacteria. Finally, the possible hosts of Barns Ness hepe-like virus 1 and 2 was considered, since these viruses were reported associated with Porifera and bivalves (Waldron, Stone & Obbard, 2018; Rosani et al., 2019), whereas they cooccur with Arthropoda (palmID) and nudibranch too.

Conclusions

A total of 20 RNA viruses were identified associated with nudibranch samples, confirming that RNA-seq datasets are useful data to untangle sample-associated viromes, particularly for under-studied taxa. While the analysis of RNA samples can reveal the active components of the biome as well as RNA viruses, DNA-based metagenomics will be required to trace the dormant DNA viruses. Starting from the evidence of RNA viruses associated with nudibranchs, future research can deepen the functional roles of viromes in these fascinating small and colourful marine species by untangling individual viromes using dedicated sample preparation approaches.

Supplemental Information

Supplemental Information 1 Combined virsorter2 and CheckV outputs.

Sequence ID, confidence value for RNA viruses, number of identified hallmark, contig length in base pair, provirus, number of identified genes, number of viral genes, number of host genes, CheckV quality value, contamination level and warnings are reported.

Click here for additional data file.

Supplemental Information 2 Taxonomic classification of nudibranch associated viruses.

For the 552 putative viral contigs, their identification by blastx, palmscan and Virsorter2 was reported together with the CAT taxonomic classification (number of annotated ORFs, superkingdom, phylum, class, order, family, genus and species.

Click here for additional data file.

Supplemental Information 3 Histogram depicting the taxonomic classification performed on the 8 samples having the highest number of viral reads.

Click here for additional data file.

Supplemental Information 4 Sequences of the 552 putative viral genomes in fasta format.

Click here for additional data file.

Additional Information and Declarations

Competing Interests

Author Contributions

Data Availability

The authors declare that they have no competing interests.

Umberto Rosani conceived and designed the experiments, performed the experiments, analyzed the data, prepared figures and/or tables, authored or reviewed drafts of the paper, and approved the final draft.

The following information was supplied regarding data availability:

All the data regarding viral sequences obtained from the metatranscriptome assembly are available in the Supplemental Files.

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
