# Peer review of "Tracing RNA viruses associated with Nudibranchia gastropods"

_PeerJ, doi:10.7717/peerj.13410_

## Round 0.1 · original submission · Major Revisions

Three reviewers have now assessed the manuscript and all agree that it is publishable, but have many suggestions for how it can be improved to ensure rigor and maximize the impact of the findings. Please pay particular attention to the the suggested changes in the Figures, widen the scope of the introduction/discussion to emphasize the value of virome exploration in underrepresented taxa, perform enhanced phylogenetic analyses of the contigs, and remove comments related to the sequencing depth coverage of reads on the viral contigs. I also agree with Reviewer #1 that "nudivirome" is problematic because the term can be confused with nudiviruses.

Reviewer 1 ·

Basic reporting

This manuscript presents a study in which the author has mined available transcriptome data in NCBI that originates from Nudibranchs, to elucidate viral diversity. This approach has been used extensively in the past elsewhere to survey RNA viral diversity in invertebrates (e.g. Shi et al 2016 Nature), but to the best of my understanding has never been applied to this charismatic group of marine invertebrates. The authors recovered a small number (31) of ‘bona fide’ viral contigs, most of which were similar to unclassified viruses. Interestingly, most identified/classified viruses were similar to a viral genome recovered from an ‘echinoderm’ (no further info available in NCBI). The author also points out considerable contamination of viromes with dinoflagellate and cnidarian genetic material.

Basic reporting is OK. Materials and methods are sufficient for another scientist to follow and repeat. I found that Fig 1, while somewhat informative, was not key to understanding the conclusions of the manuscript, so perhaps could be included as a supplemental figure. I would also promote supplemental Table 1, which includes accessions to consulted data to the main manuscript.

Experimental design

The experimental design is straightforward; the author consulted 47 metatranscriptomic databases through NCBI, assembled them, then used a couple of algorithms to highlight potential viral contigs, then used comparisons against ncbi’s databases to determine if these were ‘bona fide’. This is an appropriate pipeline as a first-order survey of novel viruses in transcriptomes. One thing that caught my eye: the authors do not perform any phylogenetic analysis of detected viruses, which would greatly benefit the manuscript. Of the 11 contigs matching Beihei picornavirus 75, it is impossible to know how closely related these contigs are. The author should perform alignment and maximum likelihood or other analysis to establish how closely related these contigs are (is it 1 across all transcriptomes? Or multiple? If the former, I’d be a bit suspicious of a contaminant).

Validity of the findings

The findings of novel viral contigs are valid. However, the discussion relating to their relative abundance is not valid in my opinion (Line 173 onwards). Just because a contig recruits a certain number of reads does not necessarily equate to abundance, since there can be un-quantified and un-expected biases introduced into the sequencing pipeline. For example, were the transcriptomes amplified by any approach? Amplification biases can lead to skewed virome data based on A+T vs G:C content and length. Were extraction protocols the same between all transcriptomes? The author mentions about 2% of reads in a library recruiting to a contig as potential evidence of significance. Viral RNAs are encapsinated and thus decay more slowly than free RNAs from hosts. Hence, the absolute representation in the library may reflect variable length of time between collection, preservation, and sequencing, and I really can’t put much stock into its virological significance. Recommend deleting this, and focusing instead on the phylogenetic analysis of new viruses detected.

Additional comments

Interesting data mining paper which elucidates new viruses from an underexplored host. Note however that even if viruses are found in host transcritpomes they may not actually infect the host, but may also be either viruses of microbes in the host microbiome or contaminants in reagents or from the surrounding environment.

As a footnote, the author should avoid "Nudivirome" as my first instinct was to see this as a paper about nudiviruses. Perhaps "Nudibranchviriome" might be more appropriate.

Reviewer 2 ·

Basic reporting

The English writing has some small issues that could be corrected.

Experimental design

Please see 4 below.

Validity of the findings

Please see 4 below.

Additional comments

The manuscript by Rosani presents an analysis of transcriptome data from nudibranchs in an attempt to uncover viral RNAs and provide insight into viruses infecting these organisms. The analyses have been carefully performed and the results are interesting and of value to the virology community, at least as a starting point to perform future targeted work on viruses in these hosts. Some comments for possible improvements to the manuscript are below.

1. It is somewhat surprising that so few transcripts could be assigned as potentially coming from DNA viruses. There seems to be only 1 such case? I believe there are quite a few known invertebrate-infecting DNA viruses, so it seems reasonable to expect some here. This could be discussed more (in the discussion section)?

2. On line 137, is “non-redundant” meant as opposed to “redundant”?

3. Table 1 - the taxonomic names should be checked for correctness - there seem to be 2 different typos in instances of Picornavirales and should “Negarviricota” be “Negarnaviricota”? Is the 1 putative DNA virus in this table or are these all RNA viruses?

4. Figure 2b would benefit from a scale to indicate the numbers of sequences the circle sizes represent - or just put the numbers into the circles.

5. Figure 3b - it is hard to make much of this with the way the virus sequences are named on the right - could they perhaps be in the same descending order as in table 1 with an indication that is so in the legend? I am assuming this is all of the sequences from the tope section of table 1? Similarly, maybe the bottom labels would be more informative as the animal species designations, or something other than the meaningless current version.

·

Basic reporting

Clarity of language: It is appreciated that the narrative is mostly written in an uncomplicated style that facilitates understanding the content. There are several unusual or non-idiomatic words or phrases, and some have been highlighted in the text. For example, line 56 “assignation” has a meaning totally different from the one the author intends, I am sure.
Literature for the content described is appropriately cited and referenced. Although the paper is self-contained, it does need a wider scope in the introduction that sets up the significance of this taxon or its virome.
Data are presented clearly and the supplementary tables are appreciated and helpful as searchable references.

Fig. 1. It is not clear that the figure is needed. If this fits into a significant question or next step, this should be justified in the text. Presenting numerical data as a figure is helpful, but is best done when that information needs to be understood by the reader to appreciate the significance of that data versus some other data set in the paper or in the field of study. Figure 1 seems like a graphical representation of numbers that are mostly record-keeping.
Fig. 2a. The tree is useful, but looks like a standard output of a software package, without a lot of guidance for the reader where the significance is. In the tree or legend, it should be made more clear which of the taxa have representation in the data analysis. This relates to the Additional comments regarding the need for more synthesis within the introduction and discussion. The names on the ends of the branches are very small.
Fig. 2b seems un-necessary. Do the circle overlaps signify shared sequences? If that is all, the list of numbers in the text conveys this well enough. In the discussion, the merits of using blastx versus the software packages was helpfully described. If this figure is there to support that comparison, it needs to be made more evident.

Supplemental Figure 1 appears to lack citation in the text. The figure appears to be a standard output of a software package. It is not clear whether this figure helps deliver the main point of the paper. The essential facts, of how many virus-like annotations were found, is easily extracted from the text.

There was a focus on a behai picorna-like virus. Please make note in the discussion that this virus-like sequence was itself assembled from a large data-set, and has no other evidence of an actual virus. While it is most likely that these viruses do exist as infectious particles, one should be accurate in all descriptions of putative virus genomes. Therefore supplementary S2 figure is not needed, and appears to be a comparison of alignments without much context. How does it carry forward the ideas about viromes?

Experimental design

This is original work that applies available bioinformatic tools to capture more knowledge from existing data sets. However it could use more context and grounding in important questions related to the study, as explained in Additional Comments.

I did not see an analysis using BLASTp or tBLASTx to search for viruses. BLASTx has a lower sensitivity to finding virus-like sequences. Virus genomes are mostly intron-free, so coding regions are rich and will give you more certainty that there is a virus there. Did you do a search or uninterrupted ORFs? Did you investigate the possibility that there would be evidence of active transposons, and how might that affect the analysis?
If the analysis did not include a translated search for matching sequences in nt/nr, it should either have one, or explain why it is not informative to do so. If the bioinformatics packages of VirSorter and checkV use protein alignments, it would be good to have that described.

Validity of the findings

The findings are valid, and the analysis conducted is a step towards understanding the virome of a lightly-studied taxon. The author does not make unsupported claims. However the findings could be more complete.
Methods are described with sufficient detail to repeat and evaluate.
The significance of the study would be greater if there were a central question or two that this study helps answer, or a future direction for research that it opens up.

Additional comments

Applying current bioinformatic tools to extract new knowledge from existing datasets is applauded. The description of putative virus genomes from metagenomic sequence data of nudibranchs is a potentially valuable resource for understanding issues related to the health and ecology of nudibranchs. The paper, in general, does a good job of describing the data used, the methods employed, and the putative virus genomes detected. However it has a deficit of synthesis and analysis within the biological, ecological or evolutionary disciplines that are relevant.
For example, the word disease appears not at all in the paper, yet viruses and disease are a major emerging problem in marine ecology, especially in invertebrates.
The introduction should include more than a passing reference to the paucity of data on nudibranch virome study. For this paper to bring the field forward, it should make the reader aware of at least one or two nudibranch- related issues. For example, is there a nudibranch mortality event that is unexplained? Are there concerns about them as invasive species, or are there concerns of diseases being moved into nudibranch habitats? Or, are the symbiotic relationships between the mollusc and other taxa important in interesting ways. How does this analysis bring us forward on an issue regarding this taxon.

---

## Round 0.2 · accepted · Accept

Thank you for carefully considering and addressing the reviewers' feedback. I especially like the use of 3 distinct techniques for identifying viruses and believe it makes the results more robust.

When you receive the page proofs, please ensure that the virus phyla names are italicized. Also a few minor typos: line 28 in revised manuscript: "cooccurrences analysis" should be "co-occurence analysis" and line 77: " diseases are not know for nudibranchs", the "know" should be "known"